# Towards a More Efficient Breast Cancer Therapy Using Active Human Cell Membrane-Coated Metal–Organic Frameworks

**DOI:** 10.3390/nano14090784

**Published:** 2024-04-30

**Authors:** Pablo Graván, Sara Rojas, Darina Francesca Picchi, Francisco Galisteo-González, Patricia Horcajada, Juan Antonio Marchal

**Affiliations:** 1Department of Applied Physics, Faculty of Science, University of Granada, 18071 Granada, Spain; gravan@ugr.es (P.G.); galisteo@ugr.es (F.G.-G.); 2Department of Human Anatomy and Embryology, Faculty of Medicine, University of Granada, 18016 Granada, Spain; 3Instituto de Investigación Biosanitaria de Granada (ibs. GRANADA), 18012 Granada, Spain; 4Biopathology and Regenerative Medicine Institute (IBIMER), Centre for Biomedical Research (CIBM), University of Granada, 18016 Granada, Spain; 5Excellence Research Unit Modelling Nature (MNat), University of Granada, 18016 Granada, Spain; 6BioFab i3D—Biofabrication and 3D (Bio)Printing Laboratory, University of Granada, 18100 Granada, Spain; 7Department of Inorganic Chemistry, Faculty of Science, University of Granada, Av. Fuentenueva s/n, 18071 Granada, Spain; srojas@ugr.es; 8Advanced Porous Materials Unit, IMDEA Energy Institute, Av. Ramón de la Sagra 3, 28935 Móstoles, Spain; darina.picchi@imdea.org; 9Escuela Internacional de Doctorado, Universidad Rey Juan Carlos, c/Tulipan, s/n, Móstoles, 28933 Madrid, Spain

**Keywords:** metal–organic frameworks, coating, human cell membrane

## Abstract

The recent description of well-defined molecular subtypes of breast cancer has led to the clinical development of a number of successful molecular targets. Particularly, triple-negative breast cancer (TNBC) is an aggressive type of breast cancer with historically poor outcomes, mainly due to the lack of effective targeted therapies. Recent progresses in materials science have demonstrated the impressive properties of metal–organic framework nanoparticles (NPs) as antitumoral drug delivery systems. Here, in a way to achieve efficient bio-interfaces with cancer cells and improve their internalization, benchmarked MIL-100(Fe) NPs were coated with cell membranes (CMs) derived from the human TNBC cell line MDA-MB-468. The prepared CMs-coated metal–organic framework (CMs_MIL-100(Fe)) showed enhanced colloidal stability, cellular uptake, and cytotoxicity in MDA-MB-468 cells compared to non-coated NPs, paving the way for these human CMs-coated MIL-100(Fe) NPs as effective targeted therapies against the challenging TNBC.

## 1. Introduction

Cancer remains a leading global cause of death [1]. Specifically, breast cancer (BC) is the most frequently diagnosed cancer in women, with over 2.3 million cases reported in 2020 alone, being the second leading cause of cancer-related deaths in women [1]. A particularly aggressive and heterogeneous form of BC is triple-negative breast cancer (TNBC), constituting 12–17% of all BC diagnoses. Associated with metastasis, drug resistance, and relapse [2], the 5-year survival rate of TNBC ranges from 8 to 16% [3]. The non-availability of specific treatment options for TNBC is usually managed through conventional chemotherapy. However, the limited solubility, poor permeability, and lack of specificity of dedicated chemotherapeutics has led to various adverse effects and reduced efficacy [4].

In the last few decades, nanomedicine has focused on overcoming TNBC treatment limitations, such as instability, low solubility, inadequate biodistribution, and potential toxicity and resistance of used drugs [5,6]. Among the available nanomedicines, biocompatible nanometric metal–organic frameworks (nanoMOFs) have gained interest in the controlled release of bioactive molecules, especially in cancer treatment [7,8,9]. Nevertheless, the current challenge for using nanoMOFs in cancer treatment resides in finely tuning their in vivo fate, recognition capabilities, and targeting. Cell membranes (CMs) surface engineering of nanoMOFs is regarded as a promising strategy to overcome these limitations. CMs retain a unique array of endogenous receptors and ligands that ensure biocompatibility and specific interactions with cells and can provide stability and protection to the encapsulated cargo, promoting progressive drug delivery, targeting, and bioavailability [10]. Initially proposed in 2016 [11], from the only 14 reports on CMs modified MOFs, mostly describing their photodynamic therapeutic abilities, only 4 examples dealt with cancer chemotherapy using the MOF/drug associations PCN-224/tirapazamine [12], ZIF-8/doxorubicin [13] or dihydroartemisinin [14], and MIL-100(Fe)/piperine [15]. Among them, the MIL-100(Fe) or [Fe_3_O(H_2_O)_2_OH(BTC)_2_] (BTC = benzene-1,3,5-tricarboxylate or trimesate) stands out due to its exceptional accessible mesoporosity (S_BET_ = 2400 m^2^·g^−1^, cages of 25 & 29 Å accessible through 5.5 & 8.6 Å apertures) [16], biostability, proven lack of in vivo toxicity, and its efficient preparation [17,18]. In this sense, only two murine CMs have been used to modify MIL-100(Fe) NPs, the first using CMs from 4T1 cells applied in ferroptosis−starvation anticancer therapy, and the other in the macrophage-membrane-vesicles from RAW 264.7 for the release of piperine [15,19], remaining unexplored the use of human CMs for the preparation of an efficient MIL-100(Fe) antitumor formulation.

Therefore, the MIL-100(Fe) NPs CMs-coating was herein originally performed using human CMs derived from the human TNBC cell line MDA-MB-468 as a unique platform to specifically internalize in cancer cells. The resulting CMs-coated MIL-100(Fe) NPs (CMs_MIL-100(Fe)) have been successfully characterized, with particular focus on their structural, chemical, and colloidal stability and their advantageously engineered surface properties. Doxorubicin (DOX) and the fluorophore (R)-(A)-4-(3-aminopyrrolidino)-7-nitro-benzofurazan (Fu) were encapsulated in the core of the systems to evaluate their efficiency in terms of cellular uptake and antitumoral activity against challenging TNBC, respectively.

## 2. Materials and Methods

### 2.1. Materials and Reagents

All reactants were commercially obtained and used without further purification. Iron (III) chloride hexahydrate (97%) and ethyl ester 1,3,5-benzenetricarboxylic (97%) were obtained from Alfa Aesar, Haverhill, MA, USA. Doxorubicin hydrochloride (Dox, 98%) was purchased from Merck, Darmstadt, Germany. All aqueous solutions were prepared using ultrapure water from a Millipore Milli-Q Academic pure-water system.

### 2.2. Experimental Techniques

Routine X-ray powder diffraction (XRPD) patterns were collected using a conventional PANalytical Empyrean powder diffractometer (PANalytical Lelyweg, Almelo, The Netherlands, *θ*–2*θ*) using *λ*Cu K*α*1, and K*α*2 radiation (*λ* = 1.54051 and 1.54433 A°). The XRPD patterns were carried out with a 2*θ* scan between 3–35° with a step size of 0.013° and a scanning speed of 0.1°·s^−1^. Fourier transform infrared (FTIR) spectroscopic analyses were performed in a Nicolet 6700 (Thermo Scientific, Waltham, MA, USA) infrared spectrometer with the help of an attenuated total reflectance (ATR) diamond accessory. High-resolution transmission electron microscopy (HRTEM) images were obtained, and energy dispersive X-ray spectroscopy (EDX) elemental mapping was performed using a high-resolution transmission electron microscope (Centre of Scientific Instrumentation of the University of Granada, Granada, Spain). 25 μL of each sample was incubated on carbon-coated grids for 5 min before being washed off with ultra-pure water. Uranyl acetate was employed for negative stained samples. Grids were observed in a High-Resolution TEM (HRTEM) TITAN from FEI Company (Hillsboro, OR, USA) operated at 300 kV. Doxorubicin and Furazan fluorescence measurements were performed with a Cary Eclipse Fluorescence Spectrometer from Agilent (Santa Clara, CA, USA) and a PerkinElmer Spectrum FL 1.4.0 (Waltham, MA, USA).

### 2.3. Synthesis of MIL-100(Fe) Nanoparticles (NPs)

MIL-100(Fe) NPs or [Fe_3_O(H_2_O)_2_OH(C_9_H_3_O_6_)_2_]·*n*H_2_O were synthesized following a microwave assisted synthesis as previously described [20]. For uptake studies, (R)-(-)-4-(3-aminopyrrolidino)-7-nitro-benzofurazan (Fu) was encapsulated into the MOF NPs based on a previous methodology [21]. In a concise procedure, 10 mg of MIL-100(Fe) NPs was suspended in 2 mL of a 32 µg·mL^−1^ Fu aqueous solution, subjected to rotational stirring for 2 h at room temperature. The resulting Fu-nanoparticles were recovered through centrifugation (13,400 rpm, 10 min), followed by washing with deionized water. The efficiency of the encapsulation process was determined to be 97%, using a fluorimeter with excitation wavelength (*λ*_exc_) set at 500 nm and emission wavelength (*λ*_em_) at 535 nm.

### 2.4. Cell Membrane Extraction

MDA-MB-468 breast adenocarcinoma CMs were isolated following a previously described method [22,23]. Briefly, cells were grown in T-175 culture flasks to full confluency and physically detached with a scrapper in PBS. Cells were collected and washed in PBS three times by centrifuging at 500× *g* for 5 min. Then, cells were suspended in a hypotonic lysis buffer consisting of 10 mM Tris-HCl pH = 7.4, 1 mM KCl, 25 mM sucrose, 1 mM MgCl_2_, and 10 μg·mL^−1^ of DNAse and RNAse and EDTA-free protease inhibitor. Cells were disrupted using a Dounce homogenizer with a tight-fitting pestle under ice-cold conditions. The solution was centrifuged at 600× *g* for 5 min. The supernatant was saved while the pellet was resuspended in a hypotonic lysis buffer and subjected to further homogenization and centrifugation. The collected supernatant was further centrifuged at 17,000× *g* for 30 min at 4 °C. The pellet was collected and washed with PBS. The final membrane-rich pellet was collected and stored for subsequent experiments. CMs were characterized by Western blot analysis. Samples were prepared at the same final protein concentration as measured by a BCA assay (Pierce). Samples were mixed with loading buffer (62.5 mM Tris-HCl (pH = 6.8 at 25 °C), 2% (*w*/*v*) sodium dodecyl sulfate (SDS), 10% glycerol, 0.01% (*w*/*v*) bromophenol blue, and 40 mM dithiothreitol (DTT)), boiled for 5 min at 100 °C. An equal sample volume was loaded into each well of a 12% gel polyacrylamide gel (12% (Mini-PROTEAN^®^ TGX™, Bio-Rad, Hercules, CA, USA). Protein was transferred to nitrocellulose membranes (Whatman, Maidstone, UK) using an XCell II Blot Module (Invitrogen, Waltham, MA, USA) in transfer buffer (Invitrogen) following the manufacturer’s instructions. Membranes were probed using a antibodies cocktail (ab140365, abcam) against Sodium Potassium ATPase, GRP78, ATP5A, GAPDH, and Histone H3 along with a horseradish peroxidase (HRP)-conjugated anti-rabbit IgG (sc-2357, Santa Cruz). Films were developed using ECL western blotting substrate (Pierce) and a Mini-Medical/90 Developer (ImageWorks, Vancouver, BC, Canada).

### 2.5. Cell Membrane Coating of MIL-100(Fe)

CM coating was carried out by mixing 1 mg of MIL-100(Fe) in 1 mL of MilliQ water previously sonicated using an ultrasound tip (Bandelin Sonoplus, Berlin, Germany, 20% amplitude, 1 min, ice), with 1 mg (protein) of membranes and sonicating the mixture for 3 min in a bath sonicator operating at 50/60 Hz and 360 W (JP Selecta™ 3000513, Abrera, Spain) [24]. After each coating procedure, CMs_MIL-100(Fe) NPs were cleaned by centrifugation (7600× *g* for 10′) to remove not coupled membranes and excess molecules, and were resuspended in phosphate buffer solution (PBS, pH = 7).

### 2.6. Physicochemical Characterization of the Prepared NPs

Hydrodynamic diameter, polydispersity index (PDI), and z-potential were determined by dynamic light scattering (DLS). Measurements were performed with a Zetasizer Nano-S system (Malvern Instruments, Malvern, UK). The self-optimization routine in the Zetasizer software (v 7.13.0.9398) was used for all measurements, and the z-potential was calculated according to the Smoluchowsky theory. Samples were diluted with a low ionic strength phosphate buffer (1.13 mM KH_2_PO_4_, pH = 7) and measured at 25 °C in triplicate. Results appeared as the mean value ± standard deviation (SD). During the colloidal stability studies, the pH effect was analyzed using buffers with identical ionic strength (0.002 M), while the ionic strength effect was analyzed at stable neutral pH of 7. The colloidal stability of the prepared systems in PBS (pH = 7.4 and 150 mM) was also determined.

### 2.7. Doxorubicin Loading Conditions

Doxorubicin (DOX) encapsulation was performed following a similar procedure as previously described [25]. 10 mg of MIL-100(Fe) NPs was suspended in 3.0 mL of DOX aqueous solution (10 mg·mL^−1^) for 24 h at room temperature (RT). The drug-loaded NPs (MIL-100(Fe)@DOX) were recovered by centrifugation (5600× *g*, 15 min, RT) and kept at 4 °C in the dark in a refrigerator. The amount of encapsulated DOX was determined by fluorescence spectroscopy (emission maximum at 551 nm when excited at 472 nm). After the DOX loading, DOX@MIL-100(Fe) NPs were coated using the same procedure as previously described for empty MIL-100(Fe). It should be noted that the amount of DOX leached during the coating process was estimated to be 0.033 wt% and considered in subsequent calculations.

### 2.8. DOX Release and Chemical Stability

The release of DOX was studied by suspending 1 mg of MIL-100(Fe)@DOX or CMs_MIL-100(Fe)@DOX in 1 mL of PBS (0.153 M, pH = 7.4). These suspensions were kept under bidimensional stirring for different incubation times (from 15 min to 5 days). At each point, an aliquot of 0.5 mL of supernatant was recovered by centrifugation (7600× *g* for 5 min) and replaced with the same volume of fresh PBS. Released DOX was quantified by fluorescence spectroscopy, and the potential linker release was also determined by HPLC. In parallel, the chemical stability of the MIL-100(Fe)@DOX was also studied through the H_3_BTC linker release by HPLC.

### 2.9. Quantification of Trimesic Acid by High Performance Liquid Chromatography (HPLC)

Quantification of trimesic acid (H_3_BTC) was performed using HPLC using a reversed phase Jasco LC-4000 series system, equipped with a PDA detector MD-4015 and a multisampler AS-4150 controlled by ChromNav software v.2 (Jasco Inc., Easton, MD, USA). A Purple ODS reverse-phase column (5 µm, 4.6 × 150 mm^2^ Análisis Vínicos, Ciudad Real, Spain) was employed. The mobile phase consisted of a 50:50 solution (*v*/*v*) of buffer (0.04 M, pH = 2.5) and methanol (MeOH). The injection volume was set at 30 µL with a flow rate of 1 mL·min^−1^ and the column temperature fixed at 25 °C. The standards used for the calibration curve consisted of trimesic acid solutions in MilliQ water solution with a concentration range from 9.65 to 0.01 µg·mL^−1^ (correlation coefficient > 0.99). The chromatogram of the standard solution showed a retention time (rt) of 2.70 min (*λ*_max_ at 225 nm).

### 2.10. Cellular Uptake

MDA-MB-468 human breast adenocarcinoma cancer cell line was obtained from American Type Culture Collection (ATCC). MDA-MB-468 cells were cultured and cultured cultured in Roswell Park Memorial Institute 1640 Medium (RPMI) (Gibco, Grand Island, NY, USA) supplemented with 10% (*v*/*v*) heat-inactivated fetal bovine serum (FBS) (Gibco), 1% L-glutamine, 2.7% sodium bicarbonate, 1% Hepes buffer, and 1% penicillin/streptomycin solution (GPS, Sigma, Kawasaki-shi, Japan). Cell were grown at 37 °C in an atmosphere containing 5% CO_2_ and 95% humidity. Cells were tested routinely for mycoplasma contamination.

Cellular uptake of the prepared NPs by MDA-MB-468 cells was assessed by flow cytometry and confocal fluorescence microscopy. For the flow cytometry assay, 1 × 10^5^ cells were seeded into 24-well culture dishes and incubated for 2, 24, and 48 h with Fu-containing coated and non-coated MIL-100(Fe) NPs at a concentration of 64 μg·cm^−2^, as previously described [26]. Then, cells were detached, centrifuged at 500 g for 5 min, washed twice with PBS, resuspended in 300 μL of PBS, and analyzed by flow cytometry with a FACS Canto II (FACSCanto II, Becton Dickinson, Franklin Lakes, NJ, USA) using the software FACSDiva 6.1.2 (Becton Dickinson) for data analysis. Confocal microscopy images were taken with a Leica Sp8 spectral laser confocal microscope (Leica, Heerbrugg, Switzerland). 3.5 × 10^4^ MDA-MB-468 cells were seeded in 35 mm glass bottom IBIDI chambers (81158, INYCOM, Zaragoza, Spain). After 24 h the culture media was changed, and cells were incubated with 64 μg·cm^−2^ of NPs for 24 h. Cells were washed twice with prewarmed (37 °C) PBS and stained with Hoechst for 5 min for nucleus visualization. Samples were washed twice with prewarmed PBS after each step. Fu-containing NPs were visualized with a 480 nm laser. During the cellular uptake assay, IBIDI chambers were placed in a thermostatic chamber, which was kept at 37 °C.

### 2.11. Cytotoxic Assays

MDA-MB-468 cells were seeded on 96 well/plates (5 × 10^3^ cells/well) and allowed to grow overnight. Free DOX, MIL-100(Fe)@DOX, and CMs_MIL-100(Fe)@DOX samples were prepared on RPMI culture medium. Cells with the corresponding incubation medium were used as positive (100%) inhibition controls, while wells with only incubation medium and no cells were used as positive (0%) growth controls. Cells were incubated for 48 h. Then, the incubation medium was removed, cells were washed with PBS, and 100 μL of 3-(4,5-dimethylthiazol-2-yl)-2,5-diphenyltetrazolium bromide (MTT, 0.6 mM) was added per well. Plates were incubated again at 37 ºC, with an atmosphere of 5% of CO_2_, and 95% humidity, for 3 h. MTT was removed, cells were washed with PBS, and 100 μL of dimethyl sulfoxide (DMSO) was added per well. Absorbance was recorded at 570 nm (HEALES MB-580 microplate reader, Araihazar, Bangladesh). Each sample was tested in triplicate.

### 2.12. Statistical Analysis and Representation

The obtained data were analyzed using Origin^®^ 2018 software (OriginLab Corporation, Northampton, MA, USA). Data appear as the mean value ± standard deviation. Data pairs were analyzed with one-way ANOVA with Tukey mean comparison method (*p* < 0.05).

## 3. Results and Discussion

### 3.1. Preparation and Physicochemical Characterization of CMs_MIL-100(Fe) NPs

The previously isolated and characterized CMs from human TNBC cells (Figure 1A) were ultrasonically suspended in an aqueous suspension of the preformed MIL-100(Fe) NPs, leading to the formation of MIL-100(Fe) coated NPs (CMs_MIL-100(Fe)). The resulting CMs-coated NPs increased their hydrodynamic diameter by 60 nm (from 214 ± 5 nm to 279 ± 16 nm) and reduced their *ζ*-potential (from −38.4 ± 1.5 mV to −27.5 ± 0.5 mV) compared to their non-coated counterpart (Figure 2A). High-resolution transmission electron microscopy (HRTEM) confirmed the nanometric size and demonstrated a visible external layer (Figure 1B) with a highly carbonaceous composition, consistent with a CM layer. Moreover, energy dispersive X-ray (EDX) spectroscopy showed differences in the Fe:U ratio (U is used here in membrane tinction) between the internal (ratio Fe:U = 5.94) and external (ratio Fe:U = 0.44) areas of the coated CMs_MIL-100(Fe) NPs, which proves the successful coating of the MOF with CMs (Appendix A). When comparing the Fourier transform infrared (FTIR) spectra of pristine and CMs-coated materials (Figure 1C), the main absorption bands of the bilayer structure (the lipid hydrocarbon tails at 2926 and 2853 cm^−1^; and the phospho- and glycosphingo-lipids, and glycosylated proteins at 1232 cm^−1^) [27] can be distinguished, confirming the presence of the CMs on the MIL-100 NPs. X-ray powder diffraction (XRPD) analysis established that the crystallinity of the MOF remains unaltered after the CM coating (Figure 1D), although the presence of the amorphous CM is also observed according to the membrane content (50.5 wt.%) in CMs_MIL-100(Fe) (Appendix A).

### 3.2. Colloidal Stability Studies

One of the advantages of CMs surface modification lies in the stabilization of the coated NPs under biological conditions. Thus, the size and *ζ*-potential of MIL-100(Fe) and CMs_MIL-100(Fe) were assessed under different pH and ionic strength biorelated conditions (Figure 2). The overall evolution of the surface charge follows a similar profile, changing from positive to negative values with increasing pH (Figure 2B,C). At acidic pH, both systems showed a positive net charge, likely related to the protonation of the carboxylic groups (pK_a_ = 3.16, 3.98, and 4.85) of MIL-100(Fe) and CMs, and the amino groups (pK_a_ = 6.5) of the CMs. On the other hand, at basic pH, carboxylate groups are negatively charged, resulting in a negative net surface potential of the system. However, the absence of amino groups in MIL-100(Fe) prevents the potential from acquiring positive values, as observed in CMs_MIL-100(Fe) NPs. The isoelectric point of MIL-100(Fe) and CMs_MIL-100(Fe) NPs is found at pH = 3.5 and 5, respectively, being associated with aggregation (Figure 2B,C). Nevertheless, both systems presented high stability within the pH range from 6 to 11, including the biological blood pH (7.4).

Similarly, colloidal stability at different ionic strength media is strongly affected by the CM coating. While the CMs_MIL-100(Fe) NPs remains stable throughout the entire range of the studied ionic strength, a drastic increase in the uncoated MIL-100(Fe) particle’s size is observed at ionic strength higher than 0.2, with a subsequent stabilization at 0.4 M (Figure 2D). When considering the surface charge of both nanosystems, a non-progressive reduction was observed with increasing ionic strength (Figure 2E). This reduction is attributed to the shielding effect of the ions in the solution, which form a layer around the particles, effectively screening the charges and decreasing their mutual repulsion forces. The absence of repulsion forces between hydrophilic MIL-100(Fe) NPs might explain their destabilization at ionic strength values > 0.2 M, while hydration forces at 0.4 M could explain their subsequent restabilization [28]. The higher colloidal stability exhibited by CMs_MIL-100(Fe) NPs could be explained by steric stabilization mechanisms due to the presence of the shell as well as strong hydration forces offered by the hydrophilic proteins and molecules of the CM coating [28,29]. Finally, both MIL-100(Fe) and CMs_MIL-100(Fe) NPs remained stable in physiological conditions (phosphate buffer saline, PBS, 0.153 M, pH = 7.4) over a period of three days, which indicates their suitability for in vivo applications (Figure 2F).

### 3.3. Cellular Uptake, Drug Release and Antitumoral Effect

CM-coated NPs inherently mimic the properties of the source cells from which their membrane is derived, bestowing a wide range of functions, such as disease-relevant targeting [30]. The specific and improved CMs_ML-100(Fe) uptake was demonstrated in human MDA-MB-468 TNBC cells. While both NPs entered the cells, even after short incubation periods (2 h) (Figure 3A), CMs_MIL-100(Fe) NPs showed significantly higher cell uptake (1.62- and 1.75-fold after 24 and 48 h, respectively) than their non-coated counterpart. Moreover, the higher uptake rate of coated nanosystems was confirmed in the confocal micrographs (Figure 3B). On the contrary, a noticeably lower fluorescence intensity in the membrane is detected in the non-coated NPs. Overall, the results indicated that the CM coating of MIL-100(Fe) NPs benefited the cellular uptake in MDA-MB-468 TNBC cells [30,31].

These findings suggest that CMs_MIL-100(Fe) could achieve an enhanced anticancer drug efficacy. Further, the antitumoral compound DOX was encapsulated in the MOF porosity, yielding a 28% drug loading by a method previously reported by our research group [25]. Subsequently, MIL-100(Fe)@DOX were coated with the CMs (CMs_MIL-100(Fe)@DOX). DOX release was evaluated under simulated physiological conditions, evidencing a drug progressive release for both CM-coated and pristine NPs (Figure 3C). When compared with the non-coated NPs, the slower and low partial release of DOX from CMs_MIL-100(Fe)@DOX might be due to the amphiphilic character of the phospholipid molecules of the lipid bilayer, which hampers the crossing of the hydrophobic DOX. Further, the MOF protective effect of the lipid bilayer was demonstrated by means of leached linker H_3_BTC, as a measure of the particle degradation, achieving a 6-fold more stable material once coated with CM (Figure 3D).

Finally, the anticancer effect of CMs_MIL-100(Fe)@DOX was demonstrated in vitro on MDA-MB-468 TNBC cells. Remarkably, CMs_MIL-100(Fe)@DOX demonstrated a higher cytotoxic effect, up to a 1.95-fold more potent antitumor effect at the highest concentration, compared to the non-coated NPs after 48 h (Figure 3E). Considering all the above, these differences could be attributed to the enhanced internalization rates of CMs coated-nanoMOF through a membrane fusion mechanism. This outcome can be attributed to the controlled and sustained release of the drug from the MIL-100(Fe) porosity.

## 4. Conclusions

In summary, these findings demonstrate that the coating with human cell membranes improves functionality of MIL-100(Fe) nanoMOFs. The successful generation of CMs_MIL-100 (Fe) NPs not only enhances their colloidal stability but also significantly improves their internalization into TNBC cells, leading to enhanced cytotoxicity. These findings mark a significant contribution to the expanding realm of biocompatible metal–organic frameworks (MOFs) and their potential for revolutionizing biomedical applications, particularly in the context of TNBC treatment.

## Figures and Tables

**Figure 1 nanomaterials-14-00784-f001:**
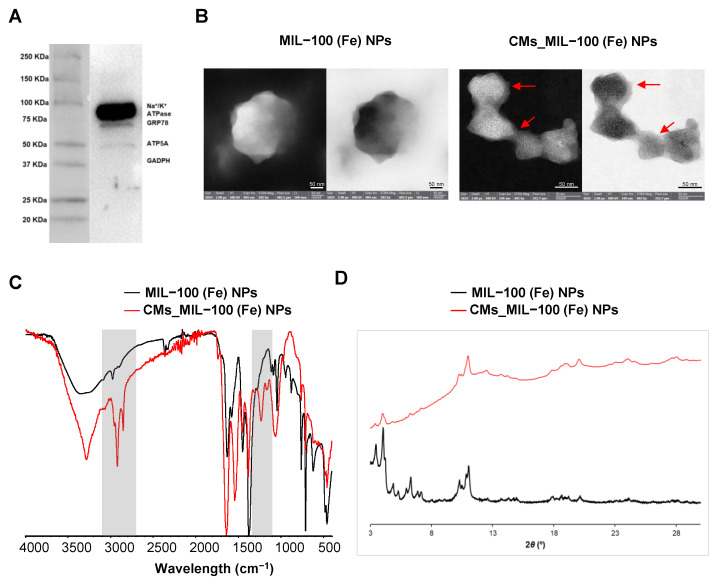
(**A**) Western-blot for a series of membrane and intracellular protein markers: plasma membrane-specific marker (Na^+^/K^+^-ATPase), endoplasmic reticulum marker (GRP78), mitochondrial marker (ATP5a), and cytosol marker (GAPDH) of the obtained CMs. The enrichment in the plasma membrane marker is observed. (**B**) HRTEM micrographs obtained in bright field mode of uncoated and CMs-coated MIL-100(Fe) NPs. The presence of CMs is highlighted by red arrows. (**C**) FTIR spectra of pristine and CMs-coated material. (**D**) XRPD patterns of CMs-coated and uncoated NPs.

**Figure 2 nanomaterials-14-00784-f002:**
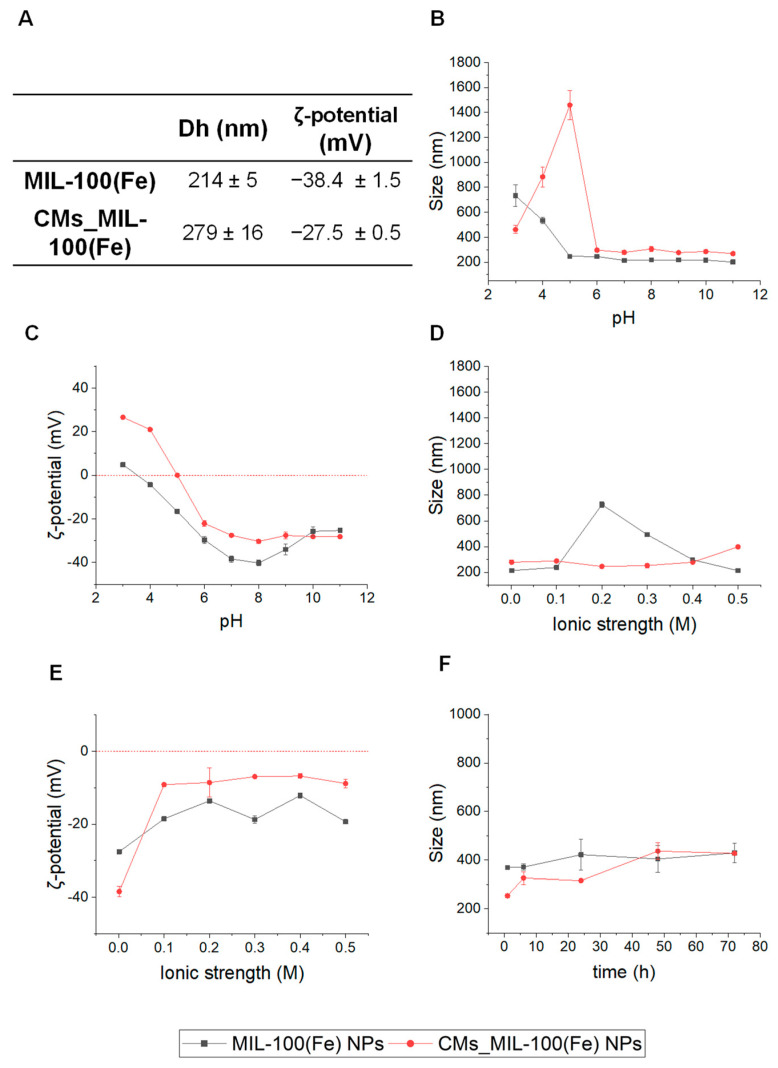
(**A**) Table presenting de Hydrodynamic diameter and surface potential of the prepared nanosystems. Evolution of (**B**) particle size, and (**C**) *ζ*-potential as a function of pH; (**D**) particle size, and (**E**) *ζ*-potential as a function of ionic strength (KNO_3_); and (**F**) particle size in PBS over time of MIL-100(Fe) (black) and CMs_MIL-100(Fe) (red) (n = 3; mean ± SD).

**Figure 3 nanomaterials-14-00784-f003:**
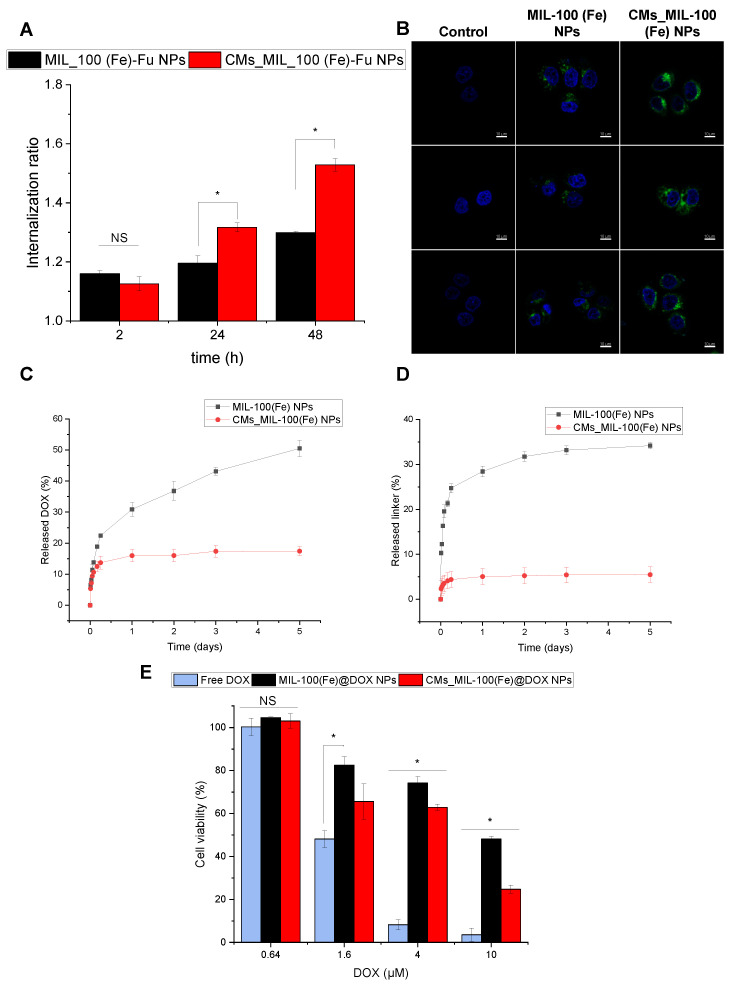
(**A**) Cellular uptake of the prepared NPs in MDA-MB-468 cells followed by flow cytometry. Data are presented as the fold-increase or internalization ratio between the median fluorescence intensity of the treated cells compared to that of the non-treated cells. (**B**) Confocal fluorescence microscopy images of MDA-MB-468 cells incubated with the prepared NPs for 24 h. Hoechst-stained nuclei and fluorescent Fu-containing NPs appear in blue and green, respectively. (**C**) Released DOX and (**D**) H_3_BTC ligand (representing MOF degradation) over 5 days from MIL-100(Fe) (black) and CMs_MIL-100(Fe) NPs (red) under simulated biological conditions. Note that lines are only visual guides. (**E**) TNBC cells viability (%) after treatment with free DOX and the prepared DOX-containing NPs for 48 h at different DOX concentrations. Statistically significant differences are highlighted with ‘*’. Analysis of variance (ANOVA) with Tukey mean comparison test (*p* < 0.05) was employed.

## Data Availability

The data that support the findings of this study are available on request from the corresponding author.

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
