# Peer review of "Towards a More Efficient Breast Cancer Therapy Using Active Human Cell Membrane-Coated Metal–Organic Frameworks"

_nanomaterials, 2024, doi:10.3390/nano14090784_

Round 1

Reviewer 1 Report

Comments and Suggestions for Authors

In this study, to achieve efficient bio-interfaces with cancer cells and improve their internalization, benchmarked MIL-100(Fe) NPs were coated with cell membranes (CMs) derived from the human TNBC cell line MDA-MB-468. The prepared CMs-coated metal-organic framework (CMs_MIL-100(Fe)) showed enhanced colloidal stability, cellular uptake, and cytotoxicity in MDA-MB-468 cells compared to non-coated NPs, paving the way for these human CMs-coated MIL-100(Fe) NPs as effective targeted therapies against the challenging TNBC. The following issues should be addressed.

1.     How about the longer term colloidal stability of the CMs_MIL-100(Fe)? Figure 2f showed the particle size in PBS over time of MIL-100(Fe) (black) and CMs_MIL-100(Fe) (red) (n = 3; mean ± SD). However, it seems that without CMs, MIL-100 also showed a stable colloidal size?

2.     In section 2.8, the DOX release was studied, however, there were no data can be found about the DOX release.

3.     In Section 2,7, the loading of DOX were studied, however, there was no related data were reported, how about the loading amount?

4.     Authors should report the coated amount of CMs.

Author Response

We are very grateful for the comments and suggestions proposed by the referee.

As requested, please find below a detailed list of revisions along with the responses to each comment by the reviewer. To facilitate the revision process, we have indicated in bold text the remarks of the referees and in black our answers. The modifications to the previous version are highlighted in green here and in the revised manuscript

Reviewer #1

In this study, to achieve efficient bio-interfaces with cancer cells and improve their internalization, benchmarked MIL-100(Fe) NPs were coated with cell membranes (CMs) derived from the human TNBC cell line MDA-MB-468. The prepared CMs-coated metal-organic framework (CMs_MIL-100(Fe)) showed enhanced colloidal stability, cellular uptake, and cytotoxicity in MDA-MB-468 cells compared to non-coated NPs, paving the way for these human CMs-coated MIL-100(Fe) NPs as effective targeted therapies against the challenging TNBC. The following issues should be addressed.

1-How about the longer term colloidal stability of the CMs_MIL-100(Fe)? Figure 2f showed the particle size in PBS over time of MIL-100(Fe) (black) and CMs_MIL-100(Fe) (red) (n = 3; mean ± SD). However, it seems that without CMs, MIL-100 also showed a stable colloidal size?

We thank the reviewer for the comment. As noted, and observed in Fig. 2F, the colloidal behavior in PBS of both nanosystems, coated and non-coated, is similar over the three-day period observed. They both remained stable and there are no significant differences between them, as stated in the text. However, we would like to highlight that while the coating does not provide a significant advantage in this specific experiment, it does enhance the system's stability under conditions of higher ionic strength. Furthermore, the coating contributes to greater uptake by tumor cells, leading to increased antitumoral activity. We hope this clarification addresses the reviewer’s concerns.

  1.    In section 2.8, the DOX release was studied, however, there were no data can be found about the DOX release.

See next comment

  1.    In Section 2,7, the loading of DOX were studied, however, there was no related data were reported, how about the loading amount?

We thank the reviewer for the comments. The conditions for DOX loading and the data on DOX release are described in Section 3.3, before discussing the antitumoral effects, and in Figure 4 of the supporting materials. However, we acknowledge that this data was not presented clearly. As also suggested by Reviewer 2, the graph depicting DOX release, previously included in the supplementary materials, is now presented as Figure 3C. We still believe that placing this data in this section, prior to discussing the antitumoral effect, is appropriate. However, we realize it was somewhat unclear. We have enhanced the clarity of the text in this section to make the data more accessible. We appreciate these comments, as they have significantly improved the clarity of the article.

  1.    Authors should report the coated amount of CMs.

We appreciate the reviewer’s comment. As described in the Materials and Methods section 2.5, to achieve the coating, 1 mg of the nanosystem is mixed with 1 mg of cell membrane proteins. This method is well-established in the field, as documented in the literature and verified by our own research (https://pubs.acs.org/doi/full/10.1021/acsami.3c13948). However, determining the exact amount of cell membranes coated on the nanosystems is notably challenging, and is out of the scope of this work. To date, there are no widely accepted methods for quantifying this, as cell membranes are complex materials and specific techniques to accurately measure them post-coating are not well-developed. However, we agree with the reviewer about the novelty of this measurement, and we don’t discard to study this in a future work.

The changes in zeta potential, as well as evidence from electron microscopy and the observed alterations in physicochemical and biological behaviors, serve as indicators of successful coating. While these changes substantiate the coating process, quantifying the exact percentage of membranes coated remains difficult. We hope this explanation addresses the reviewer's concern, and we are open to further discussion to clarify any additional points.

Reviewer 2 Report

Comments and Suggestions for Authors

This paper examines coating of metal-organic frameworks (MOFs) with human cancer cell-derived membrane for cancer therapy. The membrane-coated MOFs showed enhanced colloidal stability and cellular uptake, leading to cytotoxicity of drug-loaded MOFs. The results provide insights into cancer therapy using membrane-coated MOFs. However, some points need to be addressed.

1. Release profile of loaded drug provides important information on membrane-coated MOFs for cancer therapy. Therefore, the reviewer suggests to show Fig. S4 in the main text.

2. Relating to the comment 1, there is no description on the aim of H3BTC quantification in section 2.9. Authors need to add the description or to change the title of section 2.9 (for example, degradation of MIL-100(Fe)).

3. As for Fig. 3A, authors need to mention how to calculate the internalization ratio by flow cytometry.

Author Response

We are very grateful for the comments and suggestions proposed by the referee.

As requested, please find below a detailed list of revisions along with the responses to each comment by the reviewer. To facilitate the revision process, we have indicated in bold text the remarks of the referees and in black our answers. The modifications to the previous version are highlighted in green here and in the revised manuscript

Reviewer #2

This paper examines coating of metal-organic frameworks (MOFs) with human cancer cell-derived membrane for cancer therapy. The membrane-coated MOFs showed enhanced colloidal stability and cellular uptake, leading to cytotoxicity of drug-loaded MOFs. The results provide insights into cancer therapy using membrane-coated MOFs. However, some points need to be addressed.

Release profile of loaded drug provides important information on membrane-coated MOFs for cancer therapy. Therefore, the reviewer suggests to show Fig. S4 in the main text.

We thank the reviewer for their insightful comment. We acknowledge the importance of the drug release profile for understanding the efficacy of membrane-coated MOFs in cancer therapy. Accordingly, we have moved the graph originally shown in Fig. S4 to the main text as Fig. 3C, within Section 3.3 of the manuscript. We have also revised this section to enhance clarity. We hope these changes adequately address the reviewer’s inquiries.

  1. Relating to the comment 1, there is no description on the aim of H3BTC quantification in section 2.9. Authors need to add the description or to change the title of section 2.9 (for example, degradation of MIL-100(Fe)).

Thanks for the comment. In section 2.9 we want to highlight how the HPLC measurements were performed. The lease of H3BTC is studied during the DOX release. Now we have added some explanation in this section 2.8. Further, the title of the section was modified to “2.8. DOX release and chemical stability”.

  1. As for Fig. 3A, authors need to mention how to calculate the internalization ratio by flow cytometry.

We thank the reviewer for pointing this out. We now realize that we left out how we calculate the internalization ratio using flow cytometry. To clarify, the internalization ratio—also called the fold increase—is the ratio of the median fluorescence intensity of the treated cells to that of the untreated controls. We are adding this explanation to the figure legend of Fig. 3A to enhance clarity.